# Fault Diagnosis Method of Roadheader Bearing Based on VMD and Domain Adaptive Transfer Learning

**DOI:** 10.3390/s23115134

**Published:** 2023-05-28

**Authors:** Xiaofei Qu, Yongkang Zhang

**Affiliations:** School of Electromechanical Engineering, Guangdong University of Technology, Guangzhou 510006, China

**Keywords:** fault diagnosis, roadheader bearing, transfer learning, weak fault detection

## Abstract

The roadheader is a core piece of equipment for underground mining. The roadheader bearing, as its key component, often works under complex working conditions and bears large radial and axial forces. Its health is critical to efficient and safe underground operation. The early failure of a roadheader bearing has weak impact characteristics and is often submerged in complex and strong background noise. Therefore, a fault diagnosis strategy that combines variational mode decomposition and a domain adaptive convolutional neural network is proposed in this paper. To start with, VMD is utilized to decompose the collected vibration signals to obtain the sub-component IMF. Then, the kurtosis index of IMF is calculated, with the maximum index value chosen as the input of the neural network. A deep transfer learning strategy is introduced to solve the problem of the different distributions of vibration data for roadheader bearings under variable working conditions. This method was implemented in the actual bearing fault diagnosis of a roadheader. The experimental results indicate that the method is superior in terms of diagnostic accuracy and has practical engineering application value.

## 1. Introduction

The roadheader is a significant piece of mechanical equipment in the coal mining industry, and its main work is to carry out the tunneling work in an underground coal mine. The cutting arm of the roadheader, as its most important and key mechanism, is composed mainly of the primary cutting spindle, arm body, bearing and other primary components, which together undertake the work of cutting coal and rock. The working environment is extremely difficult and imposes strong axial and radial forces. As the main part to bear these radial and axial forces, the bearing’s health status is crucial to the roadheader. In case of failure, it can be easily stopped for maintenance, though this results in economic losses, or even production safety accidents. Therefore, the fault diagnosis of the cutting arm bearing of the roadheader is of great significance.

The characteristic signal of a cutting arm bearing fault is very weak. In the early fault stage, this signal is drowned out by complex background noise with strong impacts, and so is hard to detect. Therefore, many scholars have conducted fault diagnosis studies on roadheader bearings [1,2]. In order to study the vibration characteristics of TBM, Liu et al. [3] used the multi-body dynamics theory to establish a multi-body dynamics model of TBM, carried out a modal analysis, and drew a frequency response diagram of the system vibration characteristics, thus providing a regular basis for the load analysis of TBM. Liu et al. [4] also proposed a method of determining the residual service life of roadheader bearings based on the optimized exponential degradation model (OEDM) and used Bayesian and drift Brownian motion algorithms to implement parameter optimization and continuously update the degradation model. Experiments then verified its superiority in the operation and maintenance of key components of tunnel machinery. Additionally, Liu et al. [5] used data-driven methods to monitor and identify early faults of the cutting arm. Based on current and vibration signals, four machine learning strategies were used to study the cutting arm.

Variational mode decomposition (VMD) is a common strategy for signal decomposition in signal processing. VMD can effectively separate the various components of a signal, select the modal reconstruction signal after modal decomposition, eliminate the modes containing background noise in the signal, and then reconstruct the modes of each order, to achieve its purpose of raising a signal from background noise. As a result, this strategy is often used for signal denoising [6,7]. Bai et al. [8] have designed an intelligent fault diagnosis strategy that combines optimized VMD and migration learning for diesel engine fault diagnosis. Here, the K value of dispersion entropy is used to optimize the VMD to achieve adaptive decomposition, STFT is then used to change the signal into a two-dimensional image before, finally, the ResNet18 network is utilized for diagnosis. Their results verify the effectiveness of the proposed method in terms of noise reduction ability and diagnostic accuracy. Gharsi [9] proposed a new intelligent diagnosis system with a wavelet neural network (WNN) and non-recursive VMD to mine deeper fault features. By improving the VMD, the signal is separated and processed to reduce noise and extract fault features and the classifier is then built by WNN. In this way, it is confirmed that VMD has excellent signal processing ability compared with traditional noise reduction methods. Because VMD has excellent noise processing ability, it can be used for signal noise reduction of roadheader vibration signals.

Convolution neural networks (CNNs) have a very powerful nonlinear feature extraction ability. Through high-level abstraction, they overcome the problem of time-consuming and laborious feature extraction that is associated with traditional machine learning and incomplete feature extraction and are widely used in natural language processing [10], target tracking [11], signal recognition [12], etc. Therefore, CNN is also used in fault diagnosis and has gained excellent results [13,14]. Shao et al. [15] presented a frequency domain adaptive lightweight framework based on one-dimensional CNN with an automatic encoder function to solve the problem of minimizing domain migration through correlation alignment method (CORAL), learning domain invariant characteristics, and finally, through verification by a CRWU data set. Wang et al. [16] designed a fault diagnosis strategy based on a deformable CNN, deep short-term memory (DLSTM), and migration learning strategy to manage the issue of insufficient vibration data with labels under multiple working conditions. Here, the local features are extracted by CNN fixed geometry, and the advanced features are extracted by DLSTM. The model under one working condition is migrated and fine-tuned to another working condition via the migration learning strategy. The test results illustrate that the fault diagnosis performance of the designed model is higher than the traditional fault diagnosis performance. Therefore, CNN can be selected for feature extraction of signals.

Transfer learning (TL), as a type of deep learning, is often used to process off-design data and has achieved rich research results [17,18,19]. Wu et al. [20] constructed a TL method with fewer samples under variable working conditions (VWC) based on meta-learning for the problem of data scarcity caused by the VWC of rotating machinery and used three data sets to obtain a diagnosis with fewer samples. The results show that TL still has advantages regarding the diagnosis of rotating machinery. Cao et al. [21] sought to examine the low reusability of monitoring data for bearings under VWC, and thus proposed a feature-based multi-core balanced distribution adaptive (MBDA) TL method. This method sought to realize bearing state recognition through a stacking self-coder, and to thus obtain high accuracy. Jiang et al. [22] proposed a diagnosis method for the problem of data distribution difference (DD) under VWC. The model was validated using the bearing dataset and the gearbox experimental platform, and experiments showed the superiority of the model in terms of diagnostic accuracy and stability. Therefore, we can see that the TL method can effectively manage the issue of data DD under variable conditions.

To sum up, due to the different rock strata environments of underground coal and rock, the working conditions of the roadheader often change and are accompanied by strong axial and radial impact forces during tunneling. The collected vibration data also often have data DDs and strong background noise due to the VWC. Therefore, seeking to address the problem in which the vibration data of a roadheader under VWC have DDs and are submerged in strong background noise, this paper proposes a roadheader fault diagnosis strategy based on VMD, deep migration learning, and convolution neural network (DACNN). The vibration signal of the roadheader is decomposed using VMD, the modal reconstruction signal is then selected according to the actual situation in order to realize signal noise reduction, and the one-dimensional vibration signal after noise reduction is then input into the DACNN network model. The problem of data DD is solved by using the domain adaptive module in the network, and the convolution pooling structure with weight sharing is utilized to extract the deep features of the source domain (SD) and target domain (TD) data. Finally, the classifier carries out fault identification output and obtains the fault diagnosis result.

The remaining chapters of this research are organized as follows. In Section 2, the theoretical introduction of the method used in this research is introduced. In Section 3, the feasibility and superiority of the VMD-DACNN for fault diagnosis of a roadheader under VWC are verified by experiments. Finally, Section 4 summarizes the full text.

## 2. Theoretical Introduction

### 2.1. Variational Mode Decomposition

VMD decomposes the original data into several eigenmode functions, which can effectively suppress the endpoint effect and mode confusion through linear superposition. Compared with EMD, VMD is a linear variational mode problem, defined as a frequency-amplitude modulation signal:(1)$$u_{k}(t)=A_{k}(t)\cos(\varphi_{k}(t))$$
where $A_{k}(t)$ represents the instantaneous amplitude and $\varphi_{k}(t)$ denotes the signal phase.

By looking for K IMF components with a specific sparsity to constrain the variational model, the estimated bandwidth and sum of each component were minimized, and the sum of each component was restricted as a constraint condition to decompose the original time domain signal. First, the Hilbert transform was used to obtain the single marginal spectrum of IMF component
$u_{k}(t)$. Then, IMF central frequency
$w_{k}$ was estimated and multiplied by exponential signal
$\exp\left(jw_{k}t\right)$ to demodulate the amplitude modulation spectrum. Finally, the gradient square *L*^2^ norm of the analytic signal was given, and the variational mode was obtained as follows:
(2)$$\{\begin{array}{l} \{\underset{\left\{u_{k}\},\{w_{k}\right\}}{\min}{\left(\sum_{k=1}^{K}\left\|\partial_{t}[\left(\delta (t)+\frac{j}{\pi t}\right)*u_{k}(t)]\exp(-jw_{k}t)\right\|\right)}_{2}\\ s.t.\sum_{k=1}^{K}u_{k}(t)=f(t)\;\;\;\;\;\;\;\;\;\;\;\;\;\;\;\;\;\;\;\;\;\;\;\;\;\;\;\;\;\;\;\;\;\;\;\;\;\; \end{array}$$ where δ(t) denotes the unit impulse function, *K* represents the number of mode decomposition, *j* represents the imaginary number unit, $\partial_{t}$ is the partial derivative operation, * is the convolution operation, $\left\{u_{k}\right\}$ represents K BIMF components, and $\left\{w_{k}\right\}$ denotes the central frequency of each component.

By introducing the Lagrange multiplier λ and penalty factor α into Equation (2), the linear constraint problem is transformed into a nonlinear constraint variational problem, and the extended Lagrange is obtained:(3)$$\begin{array}{r} L(\{u_{k}\},\{w_{k}\},\lambda )=\alpha {\sum_{k=1}^{K}\left\|\partial_{t}[\left(\delta (t)+\frac{j}{\pi t}\right)*u_{k}(t)]\exp(-jw_{k}t)\right\|}_{2}^{2}\\ +{\left\|f(t)-\sum_{k=1}^{K}u_{k}(t)\right\|}_{2}^{2}+\left(\lambda (t),f(t)-\sum_{k=1}^{K}u_{k}(t)\right) \end{array}$$

For Formula (2), the alternate direction method of multipliers (ADMM) is adopted for an iterative method to seek the optimal solution, and the original data are decomposed into K IMF components. The iterative solution expression is (4)~(6):(4)$${\hat{u}}_{k}^{n+1}(w)=\frac{\hat{f}(w)-\sum_{i=1}^{k-1}{\hat{u}}_{i}^{n+1}(w)-\sum_{i=k+1}^{K}{\hat{u}}_{i}^{n}(w)+\frac{{\hat{\lambda }}_{i}(w)}{2}}{1+2\alpha {\left(w-w_{k}^{n}\right)}^{2}}$$
(5)$$w_{k}^{n+1}=\frac{\int_{0}^{\infty }w{\left|{\hat{u}}_{k}^{n+1}(w)\right|}^{2}dw}{\int_{0}^{\infty }{\left|{\hat{u}}_{k}^{n+1}(w)\right|}^{2}dw}$$
(6)$${\hat{\lambda }}^{n+1}(w)={\hat{\lambda }}^{n}(w)+\tau \left(\hat{f}(w)-\sum_{k-1}^{K}{\hat{u}}_{k}^{n+1}(w)\right)$$
where *n* represents the number of iterations, ^ denotes the Fourier transform, and τ stands for the fidelity coefficient.

IMF component center frequency and bandwidth are dynamically updated, and the expression is:
(7)$$\sum_{k=1}^{K}\left({\left\|{\hat{u}}_{k}^{n+1}(w)-{\hat{u}}_{n}^{k}(w)\right\|}_{2}^{2}/{\left\|{\hat{u}}_{k}^{n}(w)\right\|}_{2}^{2}\right)<\phi$$ where ϕ represents the discriminant accuracy, and 10^−6^ is taken in this paper.

Finally, adaptive decomposition is completed for all features in the frequency domain, and the modulation signal
${\hat{u}}_{k}^{n+1}(w)$ is decomposed into the IMF component in the time domain through Fourier transform to complete amplitude modulation and demodulation.

### 2.2. Transfer Learning

TL refers to a given SD $D_{S}=\left\{\left(x_{1}^{s},y_{1}^{s}\right),\left(x_{2}^{s},y_{2}^{s}\right),\cdots ,\left(x_{n_{s}}^{s},y_{n_{s}}^{s}\right)\right\}$ with marked information and a target domain $D_{T}=\left\{\left(x_{1}^{t}\right),\left(x_{2}^{t}\right),\cdots ,\left(x_{n_{t}}^{t}\right)\right\}$ without marked information. Here, *n_s_* and *n_t_* denote the number of data in SD and TD, respectively. The purpose of TL is to utilize the knowledge of the SD to solve the problem of the TD. The data distribution between TD $P\left(x^{t}\right)$ and SD $P\left(x^{s}\right)$ is different.

The combination of feature-based TL and neural networks is widely used in the fault diagnosis of mechanical equipment, which can handle the issue of small samples or VWC.

#### 2.2.1. Maximum Mean Difference

Maximum mean discrepancy (MMD) is the discrepancy in mapping the samples into the reproducing kernel Hilbert space (RKHS). The mean difference between the two distributions was measured in the RKHS. RKHSs are spaces that are reproducible ${\langle K(x,\cdot ),K(y,\cdot )\rangle }_{H}=K(x,y)$ and complete for the inner product of functions. The MMD distance between the two random variables is:(8)$$\sum_{k=1}^{K}\left({\left\|{\hat{u}}_{k}^{n+1}(w)-{\hat{u}}_{n}^{k}(w)\right\|}_{2}^{2}/{\left\|{\hat{u}}_{k}^{n}(w)\right\|}_{2}^{2}\right)<\phi$$ where *H* represents RKHS space, *n*_1_ and *n*_2_ represent the number of samples in X and Y. ϕ(⋅) represents the nonlinear mapping relationship between samples and RKHS. In application, to improve the measurement ability of MMDS for complex domains ϕ(⋅) is often replaced by the kernel function.

#### 2.2.2. Domain Adaptive

Domain adaptation (DA) is a popular research strategy in TL. Because of the DDs between SD and TD and the consistent classification tasks, DA uses the measurement function to measure the DDs between SD and TD and decreases the DDs in the algorithm iteration process by adding regularization in the loss function. Thus, the model learns the domain invariant characteristics between SD and TD and can then realize model migration.

The probability distribution between SD (i.e., Green) and TD (i.e., Red) can be divided into conditional distribution and marginal distribution, and the differences between the two distributions are shown in Figure 1. Edge distribution refers to the overall similarity between SD and TD. Conditional distribution refers to the similarity between each category in the two domains. According to the different DDs reduced in the process of model training, DA can be divided into edge distribution domain adaptive, conditional distribution domain adaptive, and joint distribution domain adaptive.

(1)Adaptive boundary distribution domain

The purpose of the adaptive edge distribution domain is to decrease the difference of edge probability distribution between SD and TD in model training. The marginal distribution and difference between SD and TD are calculated by
(9)$$Dis(D_{s},D_{t})\approx \left\|P(x_{s})-P(x_{t})\right\|$$

Marginal distribution domain adaptation corresponds to migration from Figure 1a to Figure 1b.

(2)The adaptive domain of conditional distribution

The purpose of the adaptive method of conditional distribution domain is to measure and reduce the difference of conditional distribution between SD and TD in the process of model training. For the conditional distribution $P(y_{s}|x_{s})$ $P(y_{t}|x_{t})$ between SD and TD, the difference is
(10)$$Dis(D_{s},D_{t})\approx \left\|P(y_{s}|x_{s})-P(y_{t}|x_{t})\right\|$$

Conditional distribution DA corresponds to the migration from Figure 1a to Figure 1c. In practice, it is rare to reduce the difference in condition distribution alone. In most methods, conditional distribution and edge distribution are measured simultaneously, that is, the joint DD between SD and TD is considered.

(3)Jointly distributed domain adaptive

The joint domain adaptive also decreases the joint DD between SD and TD, namely, conditional DD and edge DD. For the marginal distributions $P(x_{s})$ and $P(x_{t})$, and the conditional distributions $P(y_{s}|x_{s})$ $P(y_{t}|x_{t})$, the joint DD between SD and TD can be expressed as:(11)$$Dis(D_{s},D_{t})\approx \left\|P(x_{s})-P(x_{t})\right\|+\left\|P(y_{s}|x_{s})-P(y_{t}|x_{t})\right\|$$

Alignment joint DDs are shown in Figure 2.

The conditional (marginal) distributions are regarded as equally important and given the same weight in the above formula. In practice, however, the two distributions have different contributions to the alignment of SD and TD. Therefore, different weights can be assigned to the two distributions according to their contribution degrees, and thus the dynamic joint distribution domain adaptive methods can be derived.

#### 2.2.3. Convolutional Neural Network

When a CNN is dealing with image problems, the input layer is the pixel matrix of a picture; when a CNN deals with one-dimensional signal problems, the input layer is the signal column vector.

The core operation of a CNN is a convolution operation, which is characterized by a weight sharing operation. In other words, a set of filter matrices with fixed weights are used to perform inner product operations for data of different windows. If ⊙ represents the convolution operation, the convolution layer output is
(12)$$x^{l}=W^{l}⊙x^{l-1}+b^{l}$$
where $x^{l-1}$ is the output of layer L-1 and $b^{l}$ and $W^{l}$ are the bias and weight of layer *l* respectively. Figure 3 exhibits a schematic diagram of the convolution operation.

To increase the nonlinearity of the model, it should go through the nonlinear activation function $o^{l}=\sigma (x^{l})$ after the convolutional layer. Sigmoid activation function and ReLU activation function are two commonly used activation functions.

The pooling layer is another important operation in CNN, also known as down-sampling. Its function is to compress the input feature graph, simplify network complexity, reduce model parameters, and retain important information. Average pooling and maximum pooling are two common pooling operations. The purposes of the two pooling operations are to select the maximum or average value in a certain size field. With a 4 × 4 feature diagram, the maximum pooling and average pooling of a 2 × 2 pooling structure with a step length of 2 are given in Figure 4.

The fully connected layer is a layer whose matrix is flattened into a vector as input. The features extracted from the convolution pool structure can be integrated through the fully connected layer. Generally, to prevent the model from overfitting, the fully connected layer adds a dropout operation to randomly deactivate the neurons of the fully connected layer.

The output layer is also a type of fully connected layer, which is mainly the last layer of the model and outputs the classification results of the model. Typically, the model should be divided into several categories, with the output layer having a certain number of neurons. The activation function of the output layer is the Softmax activation function.
(13)yj=eaj∑i=1Neai
where $a_{j}$ represents the *j* value of the input vector of the output layer and $y_{j}$ is the label of the *j* feature.

### 2.3. VMD-DACNN Method

Because of the complex structure and VWC of the roadheader machine, the accuracy of the intelligent diagnosis model decreases. In this paper, a fault diagnosis method for a roadheader machine under VWC is proposed: VMD-DACNN.

Firstly, the VMD-DACNN model takes vibration signal as model input, SD and TD data are preprocessed by VMD, the kurtosis value is used to choose the IMF component containing the most sensitive characteristic information, and then the deep fault characteristics of the vibration signals are extracted by a multi-layer convolution pool structure. Next, the improved measurement function is used to measure the joint DD between SD and TD, and the conditional distribution alignment is improved to improve the ability of the model to extract domain-invariant features. Finally, the full connection layer integrates features and outputs the model diagnosis results.

#### 2.3.1. VMD-DACNN Networks

The VMD-DACNN model is composed of the input layer, VMD layer, multi-layer convolution pool structure, domain adaptive module, and output layer. Figure 5 gives the algorithm structure.

VMD-DACNN takes a one-dimensional vibration signal as model input. After VMD preprocessing, the vibration data of SD and TD enter the network at the same time. The deep features of SD and TD data are extracted using the convolution pooling structure with weight sharing.

In practical engineering, the collected vibration signals of a roadheader machine mostly obey Gaussian distribution, so the data distribution is determined by mean and variance. Currently, the commonly used measurement criteria MMDS can only measure the mean difference distributed in RKHS, and it is difficult for a single measurement criterion to deal with this complex domain adaptive problem. Correlated alignment can align the covariance between distributions in the feature space, and its formula is:(14)$$CORAL\left(x^{s},x^{t}\right)=\frac{1}{4d^{2}}{\left\|Cov^{s}-Cov^{t}\right\|}_{F}^{2}$$

Among them:(15)$$Cov^{s}=\frac{1}{n_{s}-1}\left(D_{S}^{T}D_{S}-\frac{1}{n_{s}}{\left(I^{T}D_{S}\right)}^{T}\left(I^{T}D_{S}\right)\right)$$
where *I* is a column vector with all entries as 1.

The method proposed in this paper combines MMD and CORAL into a new measurement function (MCMF).
(16)D(S,T)=MMD(S,T)+λCORAL(S,T)
where S,T represents samples in two respective fields and λ is the equilibrium parameter.

When MCMF is utilized to measure the joint distribution between SD and TD, it is necessary to measure the difference between the conditional (marginal) distribution between the two domains, where the marginal DD is:
(17)$$D_{1}=D\left(D_{S},D_{T}\right)$$

The conditional DD should be carried out for each category in the domain, in which the conditional DD of category c can be expressed as
(18)$$D_{2}^{c}=D\left[P\left(y^{s}=c\right)\left(f\left(x^{s}\right)|y^{s}=c\right),P\left(y^{t}=c\right)\left(f\left(x^{t}\right)|y^{t}=c\right)\right]$$
where f(·) represents the function that the model learns to extract features.

When the joint distributions of SD and TD are aligned, the model can be migrated. However, if the intra-class distributions of each category of data in the field are different, meaning that the intra-class samples are not concentrated, some samples may be correctly classified but distributed near the decision boundary. In the process of migration, the migration effect will be reduced, and the diagnostic ability of the model on the TD will be reduced. Therefore, we further improved the conditional distribution alignment. Based on the original, MCMF was utilized to measure the intra-class DD of various samples in SD and TD, and the intra-class DD of category c was:(19)$$D_{3}^{c}=\frac{1}{n_{s}(n_{s}-1)}\sum_{i\neq j}^{n_{s}}D(x_{i}^{s,c},x_{j}^{s,c})+\frac{1}{n_{t}(n_{t}-1)}\sum_{i\neq j}^{n_{t}}D(x_{i}^{t,c},x_{j}^{t,c})$$
where $x_{i}^{s,c}$ represents the *i* sample of class *c* in SD and $x_{i}^{t,c}$ denotes the *i* sample of class *c* in TD.

After VMD preprocessing and multi-layer convolution pool structure, deep features were extracted from the vibration data of the roadheader machine under VWC. Domain adaptation was carried out on the first fully connected layer, and diagnostic results were output through the output layer with Softmax as an activation function. It is noteworthy that TD data in this method do not have any label information. At this time, in the process of conditional DD, the output layer should mark false labels for TD data and update them gradually in the algorithm iteration process until the maximum number of training run has been reached.

#### 2.3.2. Fault Diagnosis

Considering the complex structure and VWC of roadheader machines and the way in which they lead to the decline of the accuracy of the intelligent diagnosis model, this chapter proposes a fault diagnosis method—VMD-DACNN—for roadheader machines under VWC. The objective function in the model training process is divided into classification loss and DA loss. Classification loss is:(20)$$l_{C}=\frac{1}{n_{s}}\sum_{i=1}^{n_{s}}J\left(T(x_{i}^{s}),y_{i}^{s}\right)$$
where T(·) is the model prediction label and J(⋅) stands for the cross-entropy loss function.

DA loss can be divided into marginal DD and improved conditional DD, and DA loss can be expressed as
(21)$$l_{DA}=D_{1}+\sum_{c}^{C}\left(D_{2}^{c}+D_{3}^{c}\right)$$

The second is the improved condition DD.

To sum up, the overall loss function of the VMD-DACNN model is
(22)$$L=l_{C}+\mu l_{DA}$$
where *μ* is the equilibrium hyperparameter. After several iterations of backpropagation, the VMD-DACNN model finally converges, thus obtaining the fault diagnosis model of a roadheader machine under VWC.

#### 2.3.3. Flowchart of the VMD-DACNN Algorithm

The detailed steps of the VMD-DACNN model fault diagnosis are proposed in this section.

(1)The vibration data of the roadheader machine collected under VWC were processed with the maximum (minimum) normalization. According to the working conditions, the data were divided into SD data with labeled information and TD data without labeled information. SD data and part of the TD data were combined to form a model training set, and the remaining TD samples were utilized as test sets to verify the model’s performance;(2)We initialized the network parameters of the VMD-DACNN model and trained the model with the training set until the maximum number of iterations was reached, leading to the fault diagnosis model of the roadheader machine under VWC;(3)The test set was input into the trained VMD-DACNN model for fault diagnosis, and the fault diagnosis results of the roadheader machine were obtained.

Figure 6 depicts the troubleshooting flowchart of the VMD-DACNN network model:

This method proposes the use of the VMD-DANN model to solve the problem of the decline of a model’s accuracy in terms of roadheader fault diagnosis that derives from the roadheader’s complex structure and changeable working conditions. This method can achieve high diagnosis accuracy of the model under different working conditions by adding Equation (21)—domain adaptation loss—to a CNN. During the training process, the target domain data do not have labeled information; however, during the testing phase, the test set samples have labels so as to evaluate the diagnostic results.

## 3. Experiment

The cutting arm bearing of the roadheader was used to verify the superiorities of the designed strategy. The test data were provided by Sanyi Heavy Industry. The high-speed end bearing type is 7217B.UA.MP FAG. The bearing data acquisition device of the roadheader is shown in Figure 7. Bearing vibration data were collected by an ICP piezoelectric acceleration sensor mounted on the side position of the roadheader bearing. The sampling frequency was 2000 Hz, and the sampling time was 10 s. The speed was divided into high and low speed with the high speed at 1482 RPM and the low speed at 736 RPM. Four kinds of state data of bearing health state, inner ring fault, outer ring fault and rolling element fault were collected at the two speeds. Samples with a length of 2048 were intercepted by sliding window, as shown in Figure 8.

VMD was applied to the designed algorithm in order to verify its feasibility, while taking the IR working conditions of the D2 data set as an example. In the VMD algorithm, the decomposition modulus k was 4, and the penalty factor α was 3000. The processing results are shown in Figure 9. Kurtosis calculation was carried out for different subcomponents, and the results are given in Table 1. It can be found that IMF1 obtains the maximum Kr value of 4.15. Therefore, these component data were selected, and other values of D1 and D2 were calculated in accordance with this method. Table 2 gives the partitioning state of the data set, where N, IR, OR and B represent the bearing health, inner ring failure, outer ring failure and rolling body failure of the roadheader, respectively. Figure 10 shows the time domain diagram of each type of data. Fifty percent of the SD and TD samples were randomly chosen to participate in the model training, and the remaining samples were utilized as test sets to test the model effect.

To verify the validity of VMD-DACNN under VWC, data sets D1 and D2 were taken as SD and TD, respectively, to form two migration tasks, D1-D2 and D2-D1. The network parameters of the VMD-DACNN model are shown in the table. The activation function of the convolutional layer and the fully connected layer of the model adopt ReLU activation function, the output layer adopts Softmax activation function, and the pooling layer adopts maximum pooling. Adam descending algorithm was used to train the model, and the number of iterations was 100. Parameters of the VMD-DACNN architecture are shown in Table 3.

Taking the D1-D2 migration task as an example, Figure 11 depicts the accuracy and loss results of the VMD-DACNN method in SD and TD, and the convergence speed of the VMD-DACNN model is fast. Even the TD sample without label information was able to converge after 40 iterations, while the accuracy of SD converges to 99.5% and TD to 95.9%. Figure 11b illustrates that, in the first 40 iterations, the loss value of VMD-DACNN decreased rapidly and then gradually converged, and that the model reached the optimal value. It should be noted that the model does indeed converge after about 40 iterations, but that we ran 100 training iterations to make the model more stable and to further improve its generalization ability. Figure 12 shows the confusion matrix of the VMD-DACNN model for D2-D1 migration tasks. It can be found that VMD-DACNN has a high classification accuracy in TD.

To highlight the superiority of the VMD-DACNN method, we combined the VMD-DACNN model proposed in this chapter with a support vector machine (SVM) and compared it with CNN; MMD-CNN, which only uses MMD as a metric function; CORAL CNN, which uses CORAL as a metric function (CORAL); and DACNN. The model network structures of several CNNs are the same. In order to decrease the instability of the model caused by randomness, 10 tests were conducted for each group. Taking the D1-D2 migration task as an example, the accuracy of each experiment and the average accuracy of eight experiments are shown in Figure 13.

Figure 13 shows that the average diagnostic accuracy of VMD-DACNN is 95.14%, and that it has the highest diagnostic accuracy. SVM is a shallow model in machine learning and has no migration module; its average accuracy is only 67.85%, which is the worst effect. CNN is a deep learning model, whose diagnostic effect is slightly better than SVM. However, because it does not have a migration module, it struggles to adapt to the fault diagnosis of roadheader bearings under VWC. MMD-CNN and CORAL-CNN are deep learning models with different measurement criteria as domain adaptive modules. The average diagnostic accuracy of the two methods is better, but there is little difference between them, with accuracy percentages of 89.79% and 88.14% respectively. DACNN is a deep domain adaptive model with an adversarial generation mechanism, and its average diagnostic accuracy is lower only than the VMD-DACNN. However, Figure 13 indicates that the diagnosis results of DACNN model are not stable. This is because the adversarial network generation needs to reach Nash equilibrium in the training process but the equilibrium standard is difficult to define. Therefore, it is easy to converge to the minimum point rather than the maximum point in the process of DACNN network training. In summary, the VMD-DACNN method can effectively reduce signal noise and further extract bearing fault characteristics and it has the strongest fault diagnosis ability and the best effect on roadheader bearings under VWC.

To verify the clustering accuracy of the model, t-sne was utilized to visualize the model results of the D2-D1 task, as shown in Figure 14. S-N, S-IR, S-OR, and S-B represent normal, inner ring fault, outer ring fault, and roller fault in the SD data set, respectively, while T-N, T-IR, T-OR, and T-B represent normal, inner ring fault, outer ring fault, and rolling body fault in the TD data set, respectively.

As shown in Figure 14, neither SVM nor CNN had a migration module, and their TD data distributions are chaotic, so the model is classified when TD does not have labels. MMD-CNN, CORAL-CNN and DACNN methods have different domain adaptive modules, which have improved the diagnostic effect in TD, but there are still some data classification errors. The VMD-DACNN model has the best classification results in the TD, small intra-class spacing, clear classification boundary, and good diagnostic effect.

The main issue studied in this manuscript is the fault diagnosis of tunnel boring machine bearings under different working conditions. The proposed method focuses on completing the migration from the source domain to the target domain without using target domain information and uses the diagnostic accuracy of the model in the target domain as the evaluation standard. The issues of small samples or of imbalanced samples were not considered in the article, but will be further explored in future research.

## 4. Conclusions

This research mainly studies the intelligent fault diagnosis strategy VMD-DACNN for a roadheader under VWC, and the conclusions are as follows. Firstly, in view of the problem that the bearing signals of a roadheader are often submerged in strong background noise and cannot be recognized, this paper adopts the VMD method to de-noise the collected fault signals and obtain the IMF component containing the optimal fault information. Then, in order to manage the complex domain adaptive problem, which is difficult to solve via common measurement criteria, a new measurement function, MCMF, was proposed in this paper. MCMF is used to simultaneously measure the mean and variance differences of distribution in feature space and to enhance the measurement ability of the measurement function for DDs. In order to manage the issue that samples are easily distributed on the decision boundary in the common conditional distribution alignment, an improved conditional distribution alignment was proposed. This proposed alignment increased the intra-class DD and reduced the intra-class distribution spacing on the basis of the original and improved the migration effect of the model. The VMD-DACNN model was verified by the measured bearing data of the roadheader. The experiments illustrate that the designed strategy has the highest classification accuracy among the existing methods. Moreover, it also has high diagnostic accuracy under VWC.

The method proposed in this paper also has some limitations that should be considered. Firstly, the sample size used in this study is relatively small, which may limit the generalizability of the results. Secondly, while transfer learning strategies are effective in addressing different distribution problems, they may not be suitable for all types of data. Thirdly, there may be other factors that contribute to bearing failures that were not considered in this study. To address these limitations and improve this work, future studies will consider using larger datasets and more diverse working conditions to validate the effectiveness and robustness of the proposed method. In addition, other machine learning techniques will be explored to complement or replace migration learning strategies. Finally, other factors that contribute to bearing failures will be further investigated, such as temperature changes or lubrication issues.

## Figures and Tables

**Figure 1 sensors-23-05134-f001:**
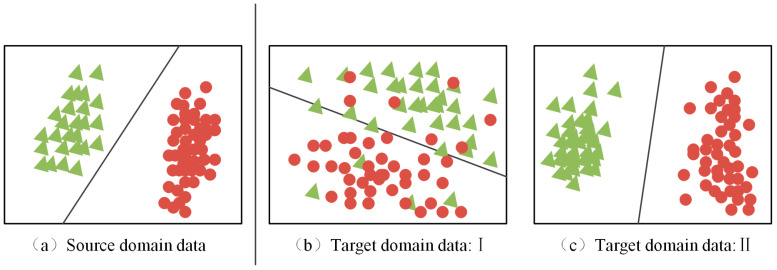
TD data with different distributions.

**Figure 2 sensors-23-05134-f002:**
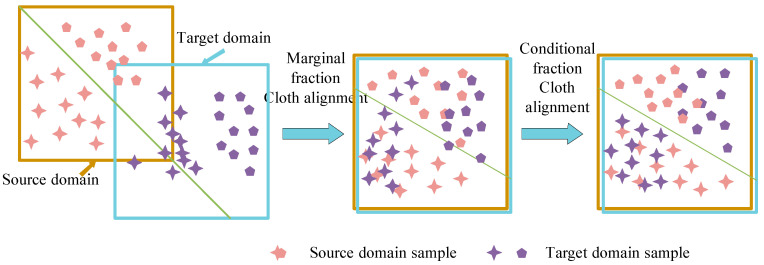
Schematic diagram showing the alignment of the joint distribution of SD and TD.

**Figure 3 sensors-23-05134-f003:**
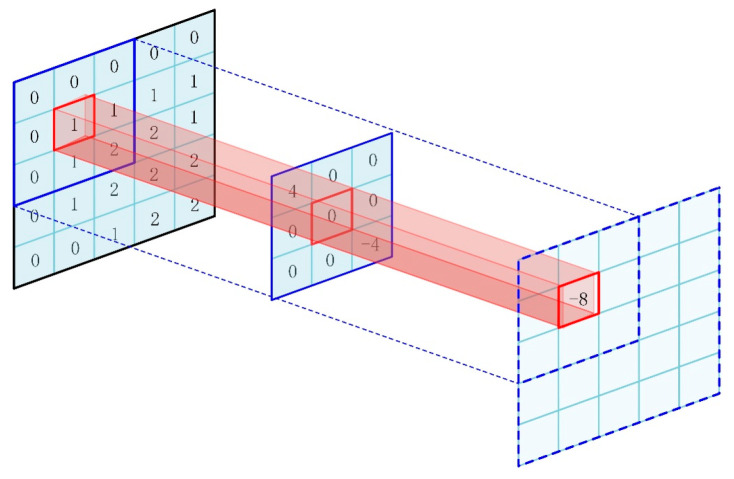
Schematic diagram of the convolution operation.

**Figure 4 sensors-23-05134-f004:**
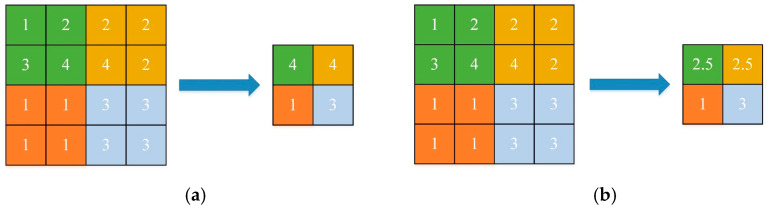
Schematic diagram of pooling operation. (**a**) Maximum pooling and (**b**) average pooling.

**Figure 5 sensors-23-05134-f005:**
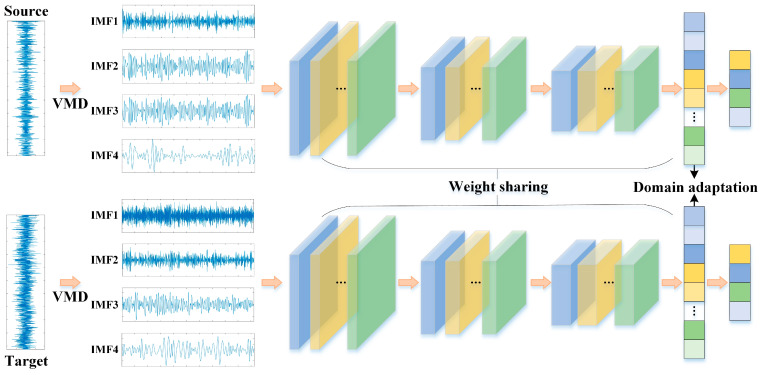
Network structure diagram of VMD-DACNN.

**Figure 6 sensors-23-05134-f006:**
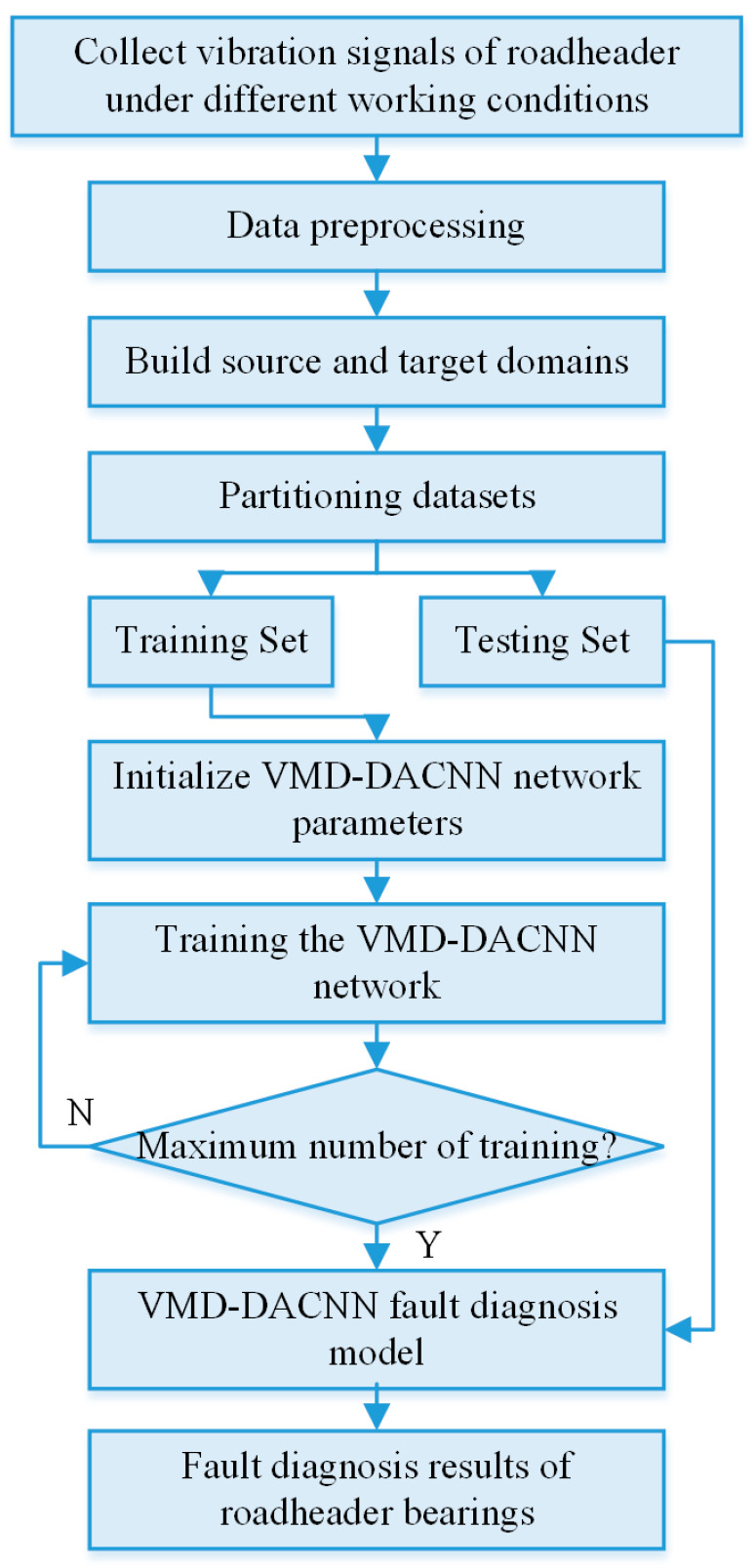
VMD-DACNN algorithm flowchart.

**Figure 7 sensors-23-05134-f007:**
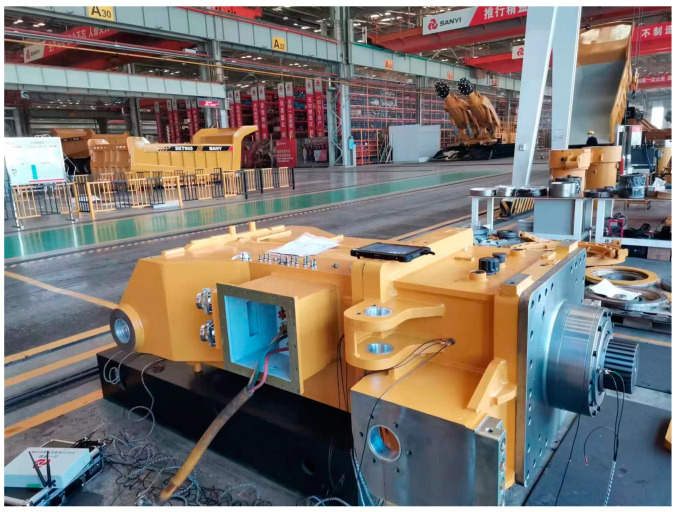
Data acquisition.

**Figure 8 sensors-23-05134-f008:**
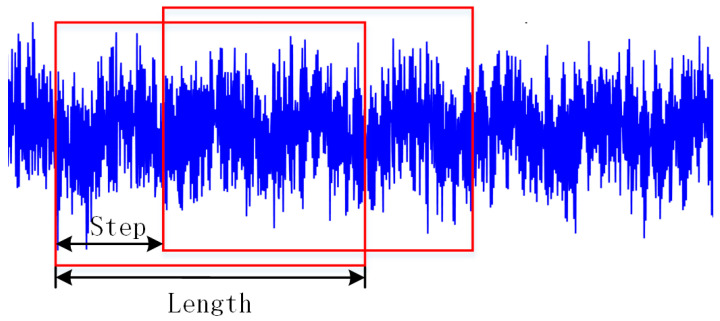
Truncation method of samples.

**Figure 9 sensors-23-05134-f009:**
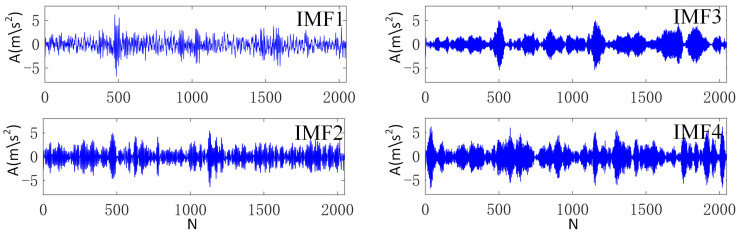
Processing result of VMD algorithm, IR in D2 data set.

**Figure 10 sensors-23-05134-f010:**
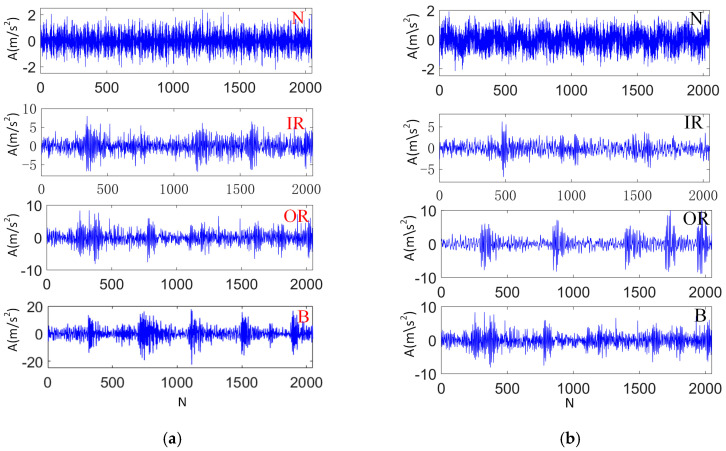
Time domain diagram of health status samples in (**a**) D1 and (**b**) D2.

**Figure 11 sensors-23-05134-f011:**
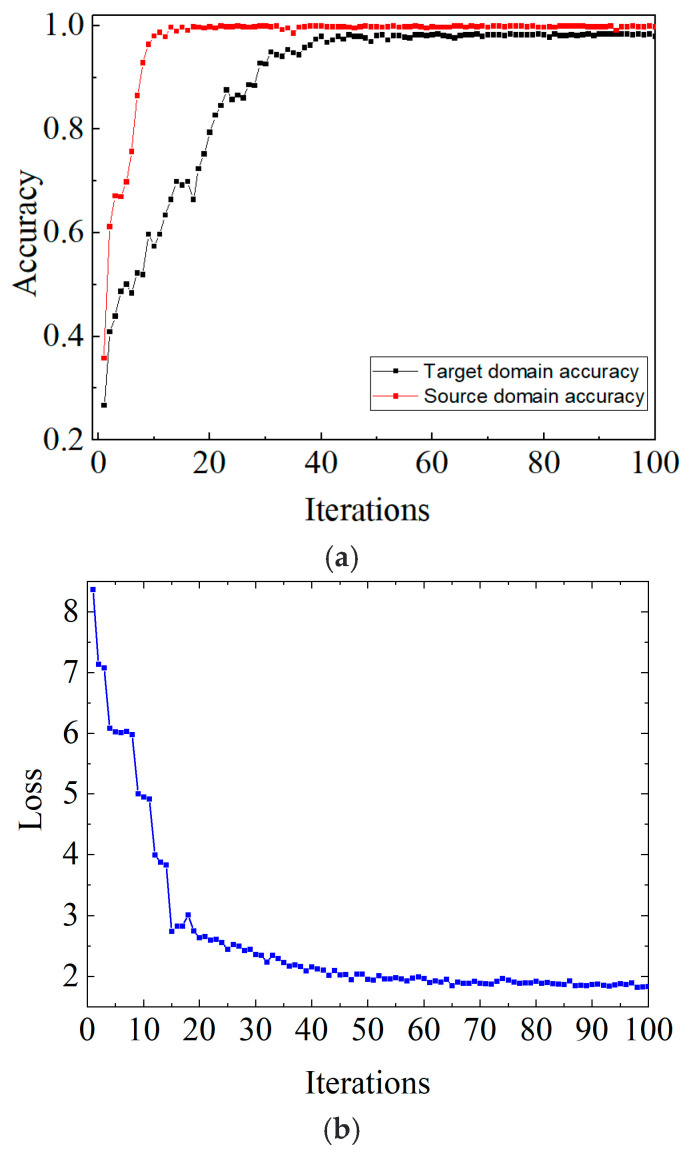
Accuracy and loss curve during training. (**a**) Accuracy of source and target domains and (**b**) loss curve.

**Figure 12 sensors-23-05134-f012:**
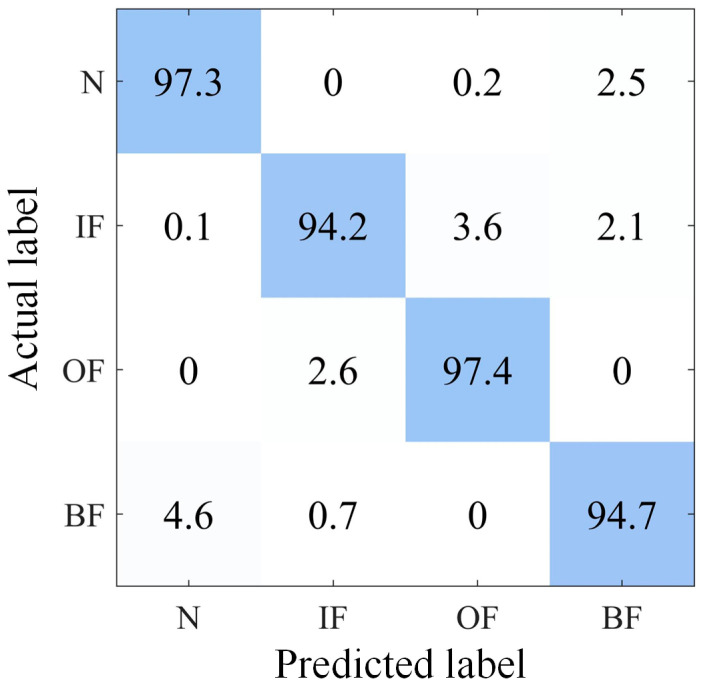
Confusion matrix of D2-D1 transfer task.

**Figure 13 sensors-23-05134-f013:**
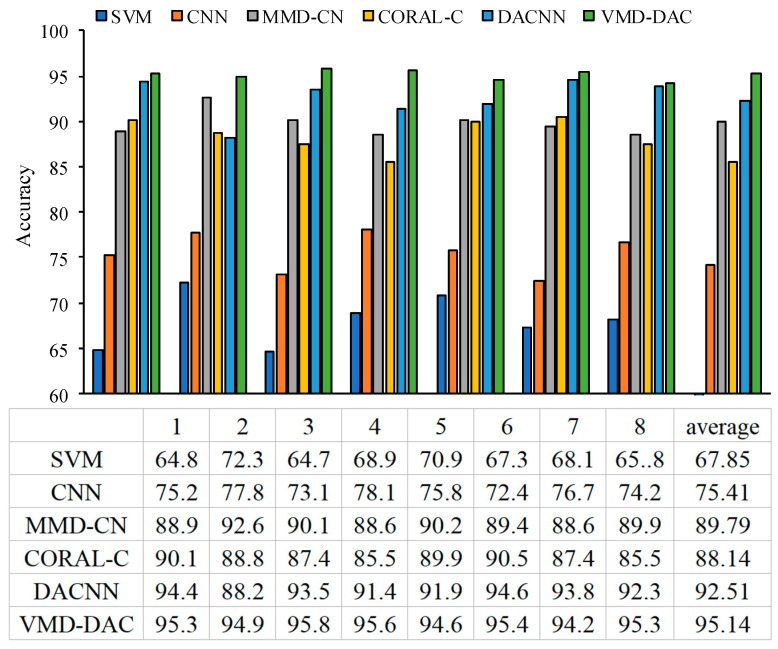
Accuracy of each algorithm in 8 experiments.

**Figure 14 sensors-23-05134-f014:**
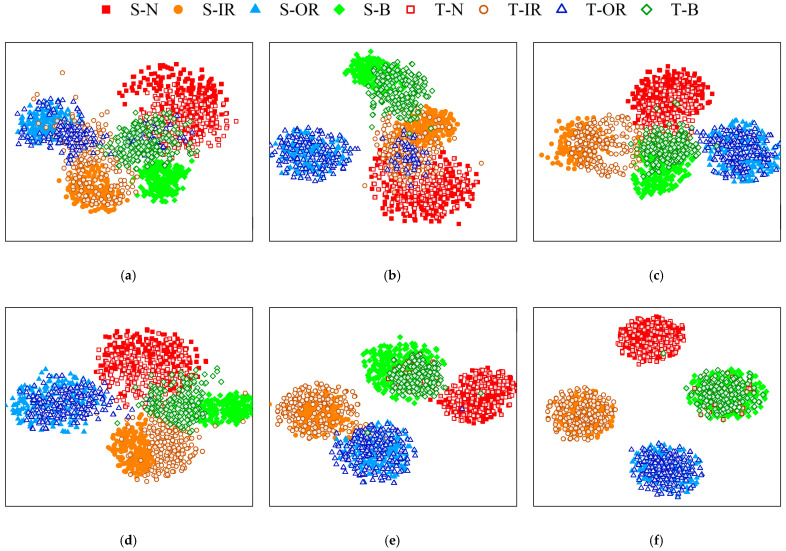
Visualization of the results of each algorithm. (**a**) SVM (**b**) CNN (**c**) MMD-CNN (**d**) CORAL-CNN (**e**) DACNN and (**f**) VMD-DACNN.

**Table 1 sensors-23-05134-t001:** Different IMF kurtosis values.

Modulus	IMF1	IMF2	IMF3	IMF4
Kr	4.15	3.11	3.913	3.22

**Table 2 sensors-23-05134-t002:** Fault type label.

Health Condition	Classify	Label	Date1 (D1) 1482 RPM	Date2 (D2) 736 RPM
Normal	0	N	400	400
Outer fault	1	OR	400	400
Inner fault	2	IR	400	400
Ball fault	3	B	400	400

**Table 3 sensors-23-05134-t003:** Parameters of the VMD-DACNN architecture.

Network Structure	Channel	Kernel Size	Fill
Input layer	/	/	/
Convolution layer 1	16	16 × 1	Yes
Normalized layer 1	/	/	/
Pooling layer 1	16	2 × 1	/
Convolution layer 2	32	3 × 1	Yes
Normalized layer 2	/	/	/
Pooling layer 2	32	2 × 1	/
Convolution layer 3	32	3 × 1	Yes
Normalized layer 3	/	/	/
Pooling layer 3	32	2 × 1	/
Fully-connected layer	256	/	/
Dropout	/	0.4	/
Output layer	4	/	/

## Data Availability

Data available on request due to restrictions.

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
