# Peer review of "Fault Diagnosis Method of Roadheader Bearing Based on VMD and Domain Adaptive Transfer Learning"

_sensors, 2023, doi:10.3390/s23115134_

Round 1

Reviewer 1 Report

In this paper, the author develops a fault diagnosis strategy combining variational mode decomposition and domain adaptive CNN. After careful reviewing, in the reviewer's opinion, the manuscript needs a minor revision before acceptance for publication in " Sensors". The topic of this paper is consistent with the aims and scope of Sensors. Thus, I recommend this paper for publication. However, a minor revision is needed.

1) Please resort the keywords according to alphabetical order.

2) During the training process of VMD-DANN model, if the target domain data does not have the label information, how can we ensure the accuracy of the model on the test set composed of the target domain?

3) 2. In Fig. 11, the model has already reached convergence after 40 iterations. What is the meaning of training 100 times? The authors should further clarify it.

4) The sample numbers and the data imbalance during the training process should be detailed. 

5) Some references seem too old to represent the research status. Please improve it. 

6) It is noted that your manuscript needs careful editing by someone with expertise in technical English editing.

Reviewer 2 Report

This paper proposes a novel fault diagnosis strategy for road-header bearings based on variational mode decomposition (VMD) and domain adaptive transfer learning. The use of VMD to decompose the collected vibration signals into sub-components and the introduction of a deep transfer learning strategy to address the problem of different distribution of vibration data under variable working conditions are contributions to the field. 

The contributions of this paper are significant. Firstly, it proposes a fault diagnosis strategy combining VMD and domain adaptive transfer learning for early detection of road-header bearing failures. Secondly, it introduces a deep transfer learning strategy to address the problem of different distribution of vibration data under variable working conditions. Thirdly, it implements the proposed method in actual bearing fault diagnosis of road-header.

The content of this paper is highly significant, as it addresses an important issue in underground mining operations - early detection of road-header bearing failures. The health of road-header bearings is critical to efficient and safe underground mining operations, and early detection of faults can prevent costly downtime and ensure worker safety. The proposed method has the potential to significantly improve the accuracy and efficiency of fault diagnosis, which can lead to improved productivity, reduced maintenance costs, and increased safety.

The authors provide a clear and concise introduction that outlines the problem statement, objectives, and contributions. The methodology section is well-organized and provides detailed information on each step of the proposed method. 

This paper will be of great interest to researchers and practitioners in the field of fault diagnosis, particularly those working in underground mining operations. The proposed method has the potential to significantly improve the accuracy and efficiency of fault diagnosis, which can lead to improved productivity, reduced maintenance costs, and increased safety. 

While the proposed method shows promising results in detecting early failures of road-header bearings, there are some limitations that should be considered. Firstly, the sample size used in this study is relatively small, which may limit the generalizability of the results. Secondly, while transfer learning strategies are effective in addressing different distribution problems, they may not be suitable for all types of data. Thirdly, there may be other factors that contribute to bearing failures that were not considered in this study.

To address these limitations and improve upon this work, future studies could consider using larger datasets with more diverse working conditions to validate the effectiveness and robustness of the proposed method. Additionally, other machine learning techniques could be explored to complement or replace transfer learning strategies. Finally, further research could investigate other factors that contribute to bearing failures such as temperature changes or lubrication issues.

It would be useful to elaborate on the background and contributions of this paper. The following papers should be added to the bibliography 

https://doi.org/10.3390/electronics12081838

https://doi.org/10.3390/electronics12081816

https://doi.org/10.3390/electronics12020409

https://doi.org/10.3390/s23073759

https://doi.org/10.3390/s23063157

https://doi.org/10.3390/s23063130

https://doi.org/10.3390/app13063413

In addition, it would be nice to add a comparison between the proposed model and existing models.

The comprehensibility of the English language utilized in the paper is satisfactory; however, there exists room for improvement in terms of the presentation and the coherence of the content to facilitate better understanding for the reader. It is suggested that enhancements be made to the structure and organization of the paper to enhance its readability and overall effectiveness.
